

# Effects of combined exercise training for adults with resistant major depression: a pilot study from the TRACE-RMD project

Mikel Tous-Espelosin[1,2], Cristóbal Pavón-Navajas[3,4,5], José Etxaniz-Oses[1,2], María Teresa Cañas-García[3,4], Sara Maldonado-Martin[1,2], Julene Orruño-Vivar[4], Pablo Corres[1,2] and Nagore Iriarte-Yoller[3,4,5]

[1] Bioaraba Health Research Institute, Physical Activity, Exercise, and Health Group, Vitoria-Gasteiz, Spain
[2] Gizartea, Kirola eta Ariketa Fisikoa Ikerkuntza Taldea (GIKAFIT), Society, Sports, and Physical Exercise Research Group, Department of Physical Education and Sport, Faculty of Education and Sport, Physical Activity and Sport Sciences Section, University of the Basque Country (UPV/EHU), Vitoria-Gasteiz, Araba/Álava, Basque Country, Spain
[3] Bioaraba Health Research Institute, New Therapies in Mental Health Group, Vitoria-Gasteiz, Spain
[4] Osakidetza Basque Health Service, Araba Mental Health Network, Psychiatric Hospital of Alava, Vitoria-Gasteiz, Araba/Álava. Basque Country, Spain
[5] Department of Medicine, Faculty of Health Sciences, University of Deusto, Bilbao, Spain

Corresponding author
Mikel Tous-Espelosin,
mikel.tous@ehu.eus

## ABSTRACT

**Background.** This pilot study analyzed the changes in body composition, cardiorespiratory fitness (CRF), biochemical profile, clinical symptoms, and quality of life (QoL) of adults with resistant major depression (RMD) following 12 weeks of combined exercise training.

**Methods.** Eighteen adults diagnosed with RMD ($42.5 \pm 9.9$ years, 66.7% women) participated in a supervised exercise group (2 days/week). The combined exercise training consisted of low-to-moderate intensity aerobic interval exercise on a bicycle and strength-resistance exercises. All variables were assessed pre- and post-intervention, including clinical symptoms with the Montgomery-Åsberg Depression Rating Scale (MADRS), the Clinical Global Impressions Scale (CGI-S), the Sheehan Disability Scale (SDS), and health-related QoL with the Short Form-36. A symptom-limited cardiopulmonary exercise peak test was performed to evaluate CRF.

**Results.** The baseline results classified the participants as metabolically unhealthy with overweight/obesity, moderate depression, and low CRF and QoL. Following the intervention, there were no significant changes ($P > 0.05$) in body composition, the main CRF physiological variables, and the biochemical profile. However, regarding clinical symptoms, the MADRS (mean difference $-8.31$, 95% confidence interval (CI) [$-15.1$ to $-1.5$], $P = 0.021$), CGI-S (mean difference -1.17, 95% CI [$-1.92$ to $-0.41$], $P = 0.006$), and SDS (mean difference -5.46, 95% CI [$-10.8$ to $-0.12$], $P = 0.046$) scores decreased, and the domains of health-related QoL –general health (mean difference 13.8, 95% CI [2.9–24.8], $P = 0.017$), vitality (mean difference 10.4, 95% CI [0.26–20.5], $P = 0.045$), social functioning (mean difference 25.9, 95% CI [8.1–43.7], $P = 0.008$), and the mental component summary (mean difference 6.9, 95% CI [1.1–12.8], $P = 0.024$) increased.

**Conclusions**. A supervised combined exercise program in people with RMD may induce positive and beneficial changes in functionality and clinical and QoL variables, and maintain body composition, CRF, and the biochemical profile. These preliminary results highlight the critical role of supervised exercise, regardless of intensity, in improving clinical symptoms in people with RMD. This study was registered with the International Standard Randomized Controlled Trial Code NCT 05136027.

# INTRODUCTION

Major depressive disorder (MDD)—a complex, multifactorial, and multigenic condition—is considered one of the most common types of depressive disorders worldwide (*Cichon et al., 2009*). Even among patients with MDD receiving adequate treatment (*i.e.*, the use of psychotherapy, a variety of antidepressants, and combined strategies of therapeutic enhancement), only 30% experience full recovery or remission (*Trivedi et al., 2006*). Patients whose depressive disorder does not respond satisfactorily to adequate treatment have harder-to-treat depression, generally referred to as resistant major depression (RMD) (*Thase & Rush, 1997*).

People with RMD generally practice an unhealthy lifestyle characterised by a lack of physical activity (PA) (*Vancampfort et al., 2017*), low cardiorespiratory fitness (CRF), tobacco use, high blood pressure, and overweight/obesity. In addition, the side effects of pharmacological treatments contribute to the appearance of different diseases, a high prevalence of metabolic and inflammatory dysregulation, as well as decreased life expectancy and quality of life (QoL) (*ten Doesschate et al., 2010*). Thus, there seems to be a bidirectional relationship, with RMD acting as both a cause and a consequence of physical inactivity (*Howland, 2010*; *Mura et al., 2014*). *Savitz et al. (2025)* recently reported that adults with depression who have high systemic levels of inflammation (defined by serum C-reactive protein (CRP) > 3 mg/L) may be at an increased level of anhedonia and worsening of clinical symptoms. Consistently, there is an evolutionary framework that links depression and chronic low-grade inflammation. It proposes that modern sedentary lifestyles diverge significantly from the PA patterns of our ancestors and contribute to increased vulnerability to mental health disorders (*Carrera-Bastos et al., 2025*). Insufficient PA plays a central role in the dysregulation of immune function, promoting systemic inflammation that is closely associated with the pathophysiology of MDD (*Schuch et al., 2016a*). Drawing on evolutionary biology, regular PA, particularly exercise aligned with basic movement patterns, might act as a powerful anti-inflammatory intervention (*Carrera-Bastos et al., 2025*).

Given this scenario, various non-pharmacological strategies have been explored as adjuncts to pharmacological treatment to enhance the prognosis and remission rates of MDD, including exercise and electroconvulsive therapy interventions (*Howland, 2010*;

*Mura et al., 2014*). According to the latest World Health Organization guidelines on PA and sedentary behaviour, recommendations for people living with a disability (including those with RMD) do not differ from those of the general elderly population (*i.e.*, a minimum of 150 min per week of moderate-to-high intensity activity or 75 min of high-to-severe intensity activity across the week, as well as at least three days a week of varied multicomponent PA that prioritise functional balance and moderate-to-higher resistance training) (*Bull et al., 2020*).

Based on a recent systematic review, in previous studies on populations with RMD, the exercise programmes consisted of 1-h sessions twice a week—typically involving aerobic exercises on a cycle ergometer or treadmill—over a period exceeding 10 weeks. These programmes have been shown to improve several parameters of depression and functioning based on the Hamilton Depression Rating Scale, the Beck Depression Inventory, the Clinical Global Impressions Scale (CGI-S), and the Global Assessment of Functioning (*Etxaniz-Oses et al., 2024*). In one study, patients in an exercise group had a 26% remission rate, compared with 0% in the control group (*Mota-Pereira et al., 2011*). Only one study incorporated a combined regimen of aerobic and resistance training (*Mather et al., 2002*). Of note, none of the cited studies controlled individualised exercise intensity.

Although general PA recommendations encourage the implementation of a moderate-to-high intensity exercise to obtain more significant health benefits, previous research has demonstrated that low-to-moderate intensity training effectively enhances various outcomes, such as CRF and body mass index (BMI), particularly in adults with chronic illnesses and a low starting level of aerobic fitness (*Horváth et al., 2022*). Moreover, in one study, it improved overall well-being and helped fill the 3-week gap that antidepressants require before showing therapeutic effects (*Schuch et al., 2016b*). Thus, low-intensity aerobic interval training (LIIT) exercise (*i.e.*, alternating loads of light-to-moderate intensity exercise) could be a useful and effective tool for people with RMD, especially those with low CRF (*De Oliveira et al., 2016*). However, to our knowledge, there is no scientific evidence regarding the effects of an exercise programme combining LIIT and resistance training in people with RMD. Therefore, we aimed to determine changes in body composition, CRF, clinical symptoms, the biochemical profile, and QoL following a 12-week combined exercise intervention for adults with RMD. We hypothesised that clinical, physical, and physiological variables concomitantly improve after the intervention.

## MATERIALS & METHODS

### Study design

The OSI Araba Research Ethics Committee (PI2021045) approved this pilot study, and all participants provided written informed consent before any data were collected. This study was registered with the International Standard Randomised Controlled Trial (Code NCT 05136027). Ethical approval for this study was granted by a local ethics committee on 7 May 2021 (Certificate No. 2021-045), in accordance with the Declaration of Helsinki. All measurements were taken before and after the 12-week intervention period, during which time the participants were involved in a supervised exercise group (Fig. 1).
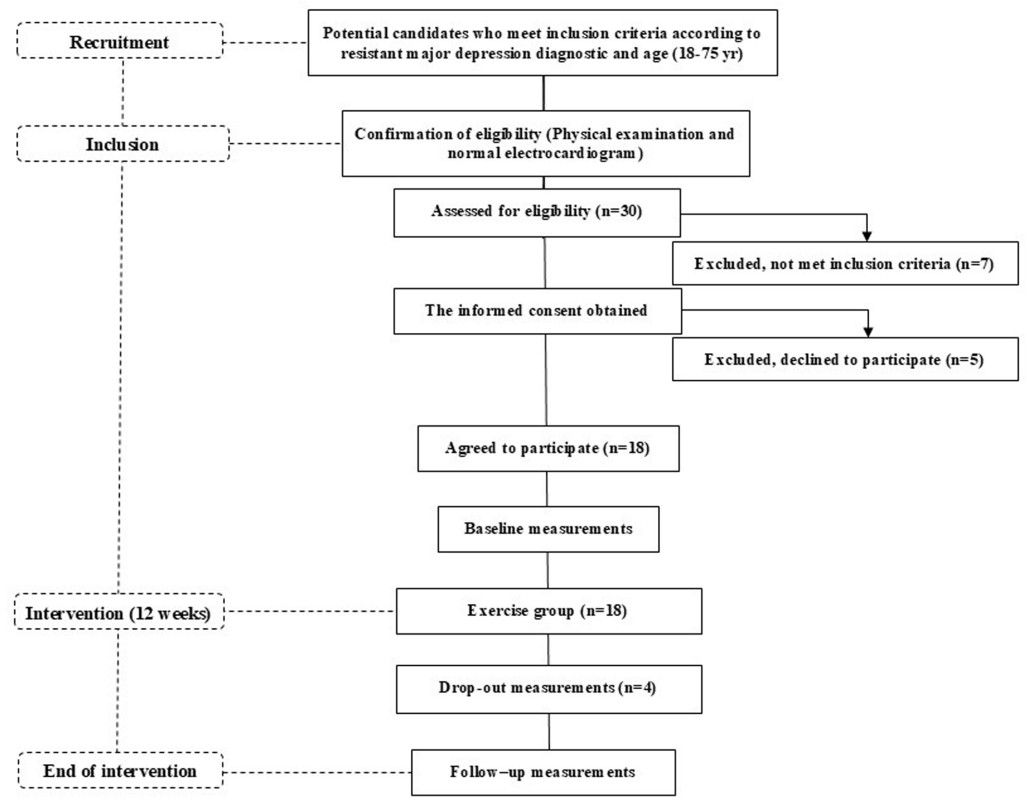

**Figure 1** **Flow diagram of the TRACE-RMD study from recruitment to the end of the intervention.**

## Study participants

Eighteen non-Hispanic, white participants (42.5 ±9.9 years old, $n = 6$ men and $n = 12$ women) participated in this study from June 2021 to May 2022. Specialised psychiatrists from the Resistant Depression Unit of the local Psychiatric Hospital recruited the participants. The participants (aged 18–75 years old) met the diagnostic criteria for RMD according to the *Diagnostic and Statistical Manual of Mental Disorders* (5th ed.). The exclusion criteria were a diagnosis of schizophrenia or other psychoses; the presence of an imminent risk of suicide; unstable or inadequately controlled medical illness (*i.e.,* an acute pathology); active substance use disorder; the presence of other comorbid psychiatric pathologies that were the primary focus of treatment; cognitive impairment determined by the Montreal Cognitive Assessment scale (<26/30); or inability to perform exercise secondary to osteoarticular, cardiovascular, or metabolic difficulties.

## Measurements

Each participant reported their age, diabetes mellitus, and cigarette smoking status. Body height and mass were measured to calculate the BMI. Waist and hip circumferences were measured in centimetres, and the waist-to-hip ratio was defined as the waist circumference divided by the hip circumference. Bioelectrical impedance analysis (BF 350; Tanita, Arlington Heights, IL, USA) was used to estimate body composition (*i.e.,* fat-free mass,

total body water, and fat mass). Blood pressure was measured by a Comfort Digital Upper Arm Blood Pressure Monitor (OMRON X3).

The Montgomery-Åsberg Depression Rating Scale (MADRS), the CGI-S, and the Sheehan Disability Scale (SDS) were used to evaluate the clinical symptoms of each participant. The MADRS is a clinical interview with extended phrased questions about symptoms of depression and anxiety, with 10 items about depression. The total score ranges from 0 (absence of depression) to 60 (a significant level of depression) (*Lobo et al., 2002*). The CGI-S has three measures: illness severity, global improvement, and an efficacy index (*Kadouri, Corruble & Falissard, 2007*). The SDS is a subjective scale used to assess both the functionality and satisfaction of treatment, with three items (social, familial, and labour), and to evaluate the incapacity due to depression (*Bobes et al., 1999*). The score from each item is summed to yield a total score that ranges from 0 to 33. A higher score indicates greater social desirability, understood as a response bias or personality trait (defensiveness) (*Sheehan, Harnett-Sheehan & Raj, 1996*).

CRF was assessed based on a symptom-limited cardiopulmonary exercise peak test performed on an electronically braked Excalibur Sport Cycle Ergometer (Lode, Groningen, the Netherlands). The exercise protocol started at 40 W with a gradual increase of 10 W per minute in a ramp protocol. Expired gas was analysed with a system (Ergo CardMedi-soft S.S., Belgium Ref. USM001 V1.0) calibrated before each test for the determination of peak oxygen consumption ($\dot{V}O_{2peak}$), defined as the highest oxygen uptake value attained towards the end of the test. Achievement of $\dot{V}O_{2peak}$ was assumed with the presence of two or more of the following four criteria: volitional fatigue (>18 on the Borg scale), a peak respiratory exchange ratio $\geq 1.1$, achieving >85% of age-predicted maximum heart rate (HR), and failure of oxygen consumption ($\dot{V}_2$) and/or HR to further increase the work rate. Ventilatory thresholds (VTs) were evaluated through standardised methodologies employing V-slope and ventilatory equivalents. The first ventilatory threshold ($VT_1$) was identified as the point of transition in the carbon dioxide production ($VCO_2$) *versus* the $VO_2$ slope from <1 to >1, or as the nadir of the ratio of ventilation (VE)/$VO_2$ *versus* the work rate relationship. The second ventilatory threshold ($VT_2$) was the nadir in the VE/$VCO_2$ ratio *versus* workload. After completing the test, the participant remained at rest on the bicycle for an additional 5 min to record recovery variables. All absolute and relative criteria for concluding the test were considered. The intensity ranges (R1 and R2) were individually tailored based on the HR to establish R1, light-to-moderate (ranging from the resting HR value to the HR at $VT_1$), and R2, moderate-to-high (ranging from the HR at $VT_1$ to the HR at $VT_2$).

A blood sample (12.5 mL) was collected from each participant in the local psychiatric hospital after an overnight fast to determine the biochemical profile, including haemoglobin, haematocrit, total cholesterol (TC), high-density lipoprotein cholesterol (HDL-C), low-density lipoprotein cholesterol (LDL-C), triglycerides (TGs), glucose, insulin, aspartate transaminase (AST), alanine transaminase (ALT), gamma-glutamyl transferase (GGT), CRP, testosterone, interleukin (IL)-1ß, IL-10, and IL-6. Homeostatic model assessment of insulin resistance (HOMA-IR) was used to evaluate insulin resistance (fasting serum insulin (µU/mL) $\times$ fasting plasma glucose (mg/dL)/405).

Health-related QoL was assessed using the Spanish version of the Short Form-36 (SF-36) questionnaire (*Alonso, Prieto & Antó, 1995*). Its items evaluate both positive and negative states of 'physical component summary' and 'mental component summary'. Eight dimensions of health are identified: physical functioning, role-physical, bodily pain, general health, vitality, social functioning, role-emotional, and mental health. For each dimension of the SF-36, the items are coded, added, and transformed into a score from 0 to 100 (with higher scores indicating higher levels of health-related QoL) using the algorithms and indications in the SF-36's scoring and interpretation manual. We received permission to use this instrument from the copyright holders.

All assessments were conducted by trained personnel following standardised protocols. Bioelectrical impedance analysis, along with all other physiological and biochemical measurements, was performed in the morning between 8:00 and 11:00 A.M., both at baseline and post-intervention, to ensure consistency and minimise circadian variation. The participants were instructed to fast for at least 8 h and to avoid vigorous PA for 24 h prior to each evaluation. Psychological and QoL measures (*e.g.,* SDS and SF-36) were administered under supervision in a quiet setting. The questionnaires used have demonstrated strong psychometric properties in clinical populations with depression. For example, the SDS has shown good internal consistency (Cronbach's alpha = 0.89), and the SF-36 has demonstrated high reliability and validity across both physical and mental health domains. These instruments were selected due to their robust validation history and relevance to the multidimensional outcomes of this study.

## Intervention

The participants participated in exercise sessions twice a week for 12 weeks under the supervision of exercise specialists at the Araba Psychiatric Hospital. All exercise sessions began and concluded with blood pressure monitoring. The training intensity was controlled by HR monitoring (Polar Electro, Kempele, Finland) and through the rate of perceived exertion using Borg's original scale (6–20 points) (*Borg, 1982*; *Scherr et al., 2013*). Each session included a 10-minute warm-up with joint mobility exercises and a 10-min cool-down period with passive stretching exercises. Combined exercise training was divided into four 10-min blocks: (1) aerobic interval exercise on the bicycle; (2) strength-resistance exercises (elastic bands, own body mass, and dumbbells); (3) aerobic interval exercise on a bicycle; and (4) pelvic girdle strengthening exercises. Power and speed were adjusted during the sessions to achieve the target HR. In the aerobic interval exercise (*i.e.,* LIIT), each participant performed a 2-min warm-up at R1, followed by six intervals (15 s at R2, followed by 1 min at R1), and finished with a 2-minute cool-down period at R1. In the resistance-circuit training protocol, each participant performed 10 strength-resistance exercises (30 s of exercise and 30 s of recovery for each exercise), including upper and lower body exercises, covering the main muscle groups, and coordinated with breathing. Additionally, after completing the circuit, each participant performed three lumbo-pelvic strength exercises (20–25 repetitions for each exercise with 30 s of recovery in between). A minimum of 85% exercise programme compliance was required for the participant to be included in the final statistical analyses.

## Statistical analysis

This pilot study had a small sample size ($n = 18$) that was not powered to detect statistical significance. Thus, the emphasis was on estimating effect sizes and 95% confidence intervals (CIs) rather than confirmatory hypothesis testing. The Shapiro–Wilk test was used to assess the normality of the baseline data. At baseline, all variables except $VT_1$, time, and distance in the cardiopulmonary test, glucose, TGs, AST, GGT, CRP, HOMA-IR, insulin, testosterone, IL-1β, IL-10, IL-6, role-physical, role-emotional, and the SDS score were normally distributed. Descriptive statistics were calculated for the baseline values for all variables. The normally distributed variables are expressed as the mean ± standard deviation (SD), and the non-normally distributed variables are presented as the median ± interquartile range (IQR) (Table 1).

Pre–post changes were analysed separately for each variable. The normality of the differences (post–pre) was assessed using the Shapiro–Wilk test. The differences for TG, CRP, HOMA-IR, insulin, testosterone, IL-1β, IL-10, and role-physical were not normally distributed, and therefore these variables were analysed using the Wilcoxon signed-rank test. The differences for all other variables were normally distributed and analysed using paired samples t-tests. For outcomes analysed with paired t-tests, the 95% CI is reported for the mean difference. For outcomes analysed with the Wilcoxon test, the 95% CI represents the median difference, assuming a symmetric distribution of the differences. Statistical significance was set at $P \leq 0.05$, although the results are interpreted as exploratory given the pilot nature of the study. All analyses were conducted using IBM SPSS Statistics Version 25.0.

## RESULTS

At baseline, the participants showed some cardiovascular risk factors (Table 1): a BMI > 25 kg/m$^2$, indicative of overweight; 34.8% fat mass, indicative of obesity; deficient CRF ($\dot{V}O_{2peak}$ = 17.3 ± 6.1 mL kg$^{-1}$ min$^{-1}$ for women and 25.7 ± 10.2 mL kg$^{-1}$ min$^{-1}$ for men); an upper optimal LDL-C concentration > 100 mg/dL; an atherogenic index (TC/HDL-C) > 3.5; fasting plasma glucose > 100 mg/dL; and a proinflammatory status with CRP > 3 mg/L and IL-1ß < 0.5 pg/mL. Conversely, the participants had normal blood pressure (123/79 mmHg), HDL-C (>40 mg/dL), TC (<200 mg/dL), TGs (<200 mg/dL), insulin (<16.7 µU/mL), HOMA-IR (<3.8), liver enzymes (AST <30 U/L, ALT <30 U/L, and GGT <50 U/L), testosterone (0.2 (0.02) ng/dL for women and 3.8 (1.3) ng/dL for men), IL-10 (<7 pg/mL), and IL 6 (<9 pg/mL). Overall, at baseline the participants were metabolically unhealthy with overweight/obesity according to the Wildman-modified criteria, considering glucose and CRP levels (*Martínez-Larrad et al., 2014*), and had low CRF.

Table 2 presents the psychiatric and QoL characteristics at baseline. The participants had moderate depression based on the MADRS (29.1) and were moderately ill based on the CGI-S (4.2). The SDS score (25.0 ± 6.0) indicated high disability (83.3%), considering that a value of 30 represents a 100% impact on functionality. Furthermore, the participants had low QoL values for the physical (<50) and mental (<50) component summary analyses

**Table 1  Physical, physiological, and biochemical characteristics at baseline from TRACE participants ($N = 18$).**

| | |
|---|---|
| Sex (men/women) | 6/12 |
| Age (yrs) | $42.5 \pm 9.9$ |
| Cigarette smoking (%) | 22.2 |
| Blood pressure (mmHg) | 123/79 |
| **Body Composition** | |
| Body mass (kg) | $78.0 \pm 18.2$ |
| BMI (kg m$^{-2}$) | $29.2 \pm 4.9$ |
| Waist/hip ratio | $0.92 \pm 0.09$ |
| FFM (%) | $65.1 \pm 9.6$ |
| FBM (%) | $34.8 \pm 9.6$ |
| **Submaximal and Peak Physiological Variables** | |
| $VT_1$ (mL kg$^{-1}$ min$^{-1}$) | $13.0 \pm 6.0$ |
| $HR_{peak}$ (bpm) | $139.1 \pm 29.9$ |
| $\dot{V}O_{2peak}$ (L min$^{-1}$) | $1.60 \pm 0.80$ |
| $\dot{V}O_{2peak}$ (mL kg$^1$ min$^{-1}$) | $20.1 \pm 8.4$ |
| $MET_{peak}$ | $5.8 \pm 2.4$ |
| $RER_{peak}$ | $1.04 \pm 0.12$ |
| **Biochemical Variables** | |
| Hemoglobin (mg/dL) | $14.1 \pm 1.5$ |
| Hematocrit (%) | $42.7 \pm 4.7$ |
| Glucose (mg/dL) | $101.9 \pm 45.3$ |
| HOMA-IR | $2.8 \pm 3.0$ |
| Insulin (µU/mL) | $10.4 \pm 8.5$ |
| TC (mg/dL) | $186.7 \pm 53.2$ |
| HDL-C (mg/dL) | $48.6 \pm 12.8$ |
| LDL-C (mg/dL) | $127.2 \pm 39.8$ |
| TC/HDL-C ratio | $4.0 \pm 1.3$ |
| TG (mg/dL) | $101.5 \pm 130.0$ |
| AST (U/L) | $18.0 \pm 6.0$ |
| ALT (U/L) | $22.4 \pm 10.2$ |
| GGT (U/L) | $23.0 \pm 20.0$ |
| CRP (mg/L) | $4.4 \pm 5.5$ |
| Testosterone (ng/dL) | $0.2 \pm 0.7$ |
| IL 1 beta (pg/ml) | $0.7 \pm 2.1$ |
| IL 10 (pg/ml) | $1.6 \pm 0.0$ |
| IL 6 (pg/ml) | $4.7 \pm 6.0$ |

Notes.

ALT, alanine transaminase; AST, aspartate transaminase; BMI, body mass index; CRP, C-reactive protein; FFM, fat-free mass; FBM, fat body mass; GGT, gamma-glutamyl transferase; HDL-C, high-density lipoprotein cholesterol; HOMA-IR, homeostasis model assessment of insulin resistance; HR, heart rate; IL, interleukin; LDL-C, low-density lipoprotein cholesterol; $MET_{peak}$, peak metabolic equivalent of task; $RER_{peak}$, peak respiratory exchange ratio; TC, total cholesterol; TG, triglycerides; $\dot{V}O_{2peak}$, peak oxygen uptake; VT, ventilatory threshold.

**Table 2** Psychiatric, medication and Health-related Quality of Life characteristics at baseline from TRACE participants ($N = 18$).

| Clinical symptoms variables | |
| --- | --- |
| MADRS | $29.1 \pm 12.2$ |
| CGI-G | $4.2 \pm 1.3$ |
| SDS | $25.0 \pm 6.0$ |
| **SF-36 questionnaire** | |
| Physical functioning | $61.1 \pm 26.9$ |
| Role-physical | $50.0 \pm 81.0$ |
| Bodily pain | $60.7 \pm 30.9$ |
| General health | $28.7 \pm 19.6$ |
| Vitality | $26.4 \pm 16.2$ |
| Social functioning | $39.6 \pm 33.8$ |
| Role-emotional | $22.2 \pm 36.2$ |
| Mental health | $40.9 \pm 22.1$ |
| Physical component summary | $42.4 \pm 12.3$ |
| Mental component summary | $28.3 \pm 12.4$ |
| **MEDICATION** | |
| Antidepressant | |
| 1 (%) | 41.2 |
| 2 (%) | 41.2 |
| ≥3 (%) | 11.8 |
| Antipsychotic (%) | 29.4 |
| Stabilizers (%) | 58.8 |

**Notes.**
MADRS, Montgomery-Åsberg Depression Rating Scale; CGI-G, Clinical global impression scale; SDS, Sheehan Disability Scale.

of the SF-36. Most participants were taking one (41.2%) or two (41.2%) antidepressants (Table 2) combined with antipsychotics (29.4%) and stabilisers (58.8%).

After the intervention, four participants dropped out for logistical (not clinical) reasons. Therefore, the total sample for statistical analysis after the intervention was 14. There were no significant changes ($P > 0.05$) in the body composition variables, submaximal and peak physiological variables, or the biochemical profile, except for haemoglobin (mean difference = 0.64 mg/dL, 95% CI [0.18–1.10] mg/dL, $P = 0.010$,), haematocrit (mean difference = 2.21 mg/dL, 95% CI [0.92–3.51] mg/dL, $P = 0.003$), and AST (mean difference = 3.07 mg/dL, 95% CI [0.62–5.52] mg/dL, $P = 0.018$). After the intervention, there was a significant reduction in the clinical symptoms of depression based on the MADRS (mean difference = $-8.31$, 95% CI [$-15.1$ to $-1.5$], $P = 0.021$), the CGI-S (mean difference = $-1.17$, 95% CI [$-1.92$ to $-0.41$], $P = 0.006$), and the SDS (mean difference = $-5.46$, 95% CI [$-10.8$ to $-0.12$], $P = 0.046$) (Fig. 2). Regarding changes in the domains of health-related QoL (Fig. 3), after the intervention there was an increase in general health (mean difference = 13.8, 95% CI [2.9–24.8], $P = 0.017$), vitality (mean difference = 10.4, 95% CI [0.26–20.5], $P = 0.045$), social functioning (mean difference = 25.9, 95% CI

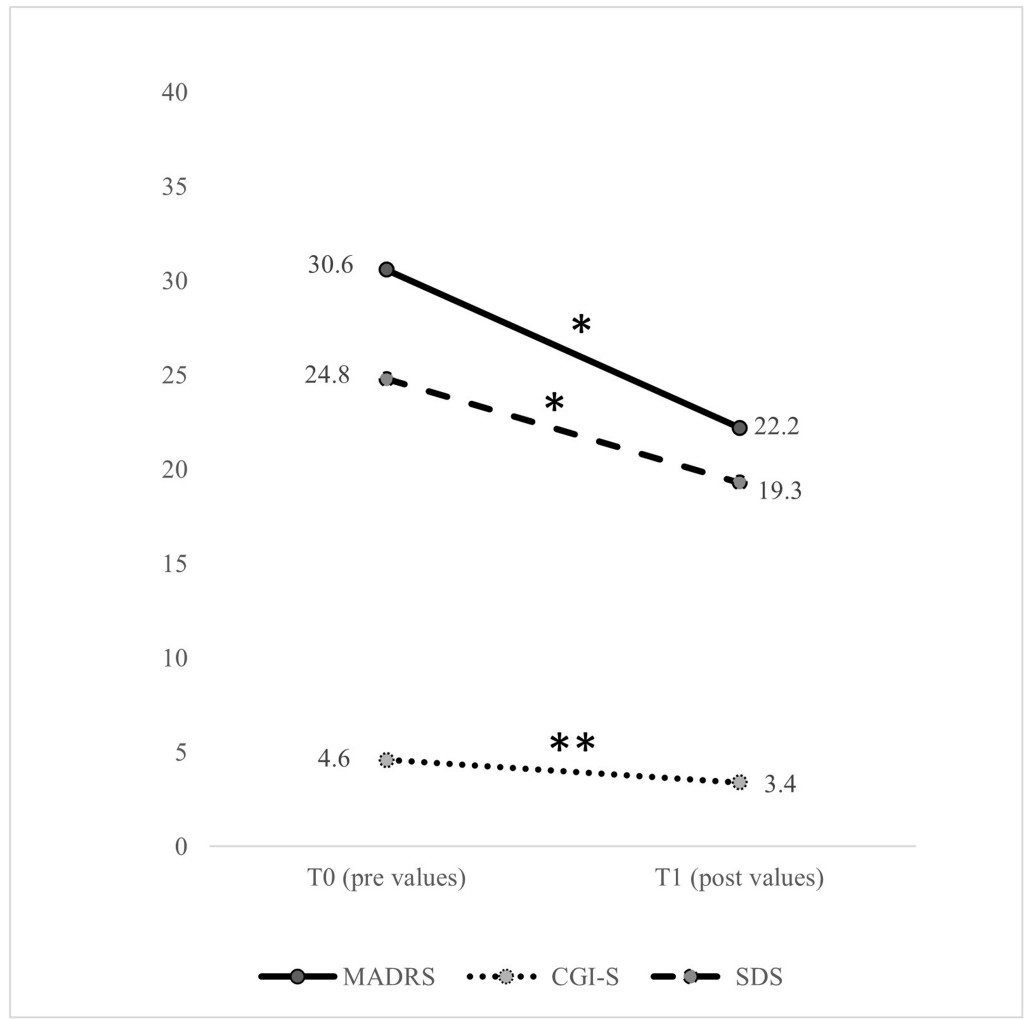

**Figure 2** **Pre- and post-intervention in clinical symptoms variables.** CGI-S, Clinical Global Impressions Scale; MADRS, Montgomery-Åsberg Depression Rating Scale; SDS, Sheehan Disability Scale; T0, pre values; T1, post values. The significant difference between T0 and T1 values was set at $P < 0.05$. *$P < 0.05$, **$P < 0.01$.

[8.1–43.7], $P = 0.008$), and the mental component summary (mean difference $= 6.9$, 95% CI [1.1–12.8], $P = 0.024$).

## DISCUSSION

The present pilot study is unique in that it investigated the effects of a supervised, combined exercise training programme (LIIT + resistance exercise training) in adults diagnosed with RMD. Based on the preliminary findings, the combined exercise programme induced positive and beneficial changes in functionality, clinical symptoms (based on the MADRS, CGI-S, and SDS), and QoL. However, there were no associated changes in

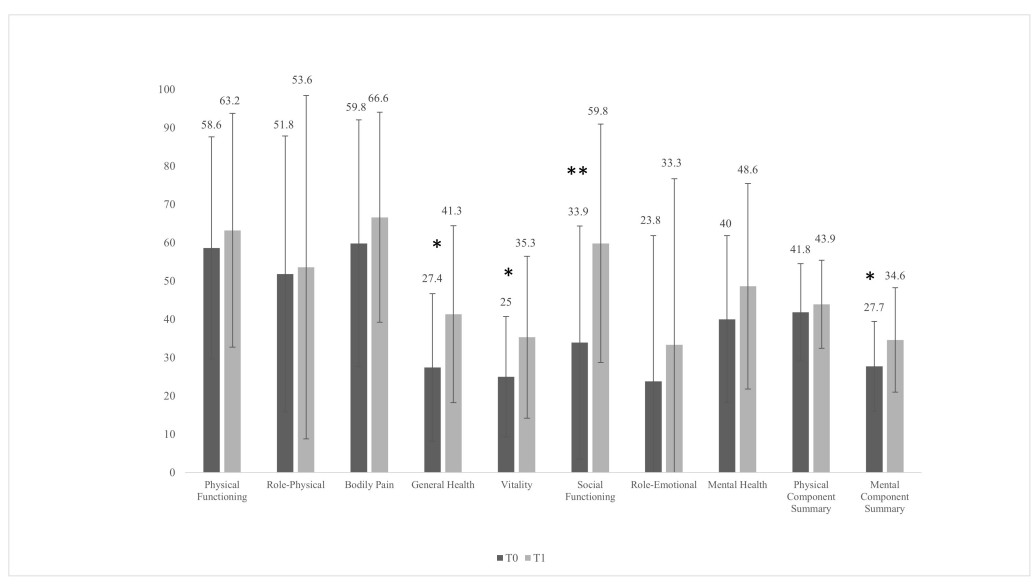

**Figure 3** **Pre- and post-intervention health-related quality of life variables with the short form-36 questionnaire.** T0, pre values; T1, post values. The significant difference between T0 and T1 values was set at $P < 0.05$. *$P < 0.05$, **$P < 0.01$.

body composition, submaximal and peak physiological exercise-related variables, or some of the analysed biochemical variables.

For patients with RMD, exercise interventions have been considered a non-pharmacological treatment to help improve prognosis and remission rates of MDD (*Howland, 2010*; *Mura et al., 2014*). In the present pilot study, the participants showed improvement in all assessed depression and functioning parameters (Figs. 2 and 3). These results are consistent with previous reports concerning the positive effects of exercise on patients with RMD (*Mota-Pereira et al., 2011*) and MDD (*Trivedi et al., 2006*). Treatment with antidepressants has been shown to increase the volume of the hippocampus, anterior cingulate, and orbitofrontal cortex; to improve white matter integrity; and to alter the functional dynamics of frontal-limbic neural networks in adults with depression (*Singh & Gotlib, 2014*). However, growing evidence suggests that both exercise and antidepressant use may relieve depression through similar neuromolecular mechanisms (*Garza et al., 2004*; *Hird et al., 2024*). Increased expression of neurotrophic factors, such as brain-derived neurotrophic factor (*Garza et al., 2004*), greater availability of serotonin and norepinephrine, regulation of hypothalamic–pituitary–adrenal axis activity, reduced systemic inflammatory signalling (*Mathur & Pedersen, 2008*), and enhanced dopamine transmission can be achieved with PA and exercise interventions (*Hird et al., 2024*). These processes help develop new neurons, enhance synaptic connections, and increase cerebral vasculature and hippocampal plasticity (*Sun et al., 2022*). Hence, once can assume that if clinical symptoms improve, there is a reciprocal interaction with improvements in psychosocial functioning and QoL (*Papakostas et al., 2004*). After the 12-week intervention, the participants showed an increase in general health, vitality, social functioning, and mental

component summary scores (Fig. 2). Although a previous study implementing a 12-week walking programme (30–45 min/day) suggested it may not be sufficient to significantly improve several domains of QoL (except for the physical functioning domain) in patients with RMD (*Mota-Pereira et al., 2011*), another more recent study found improvements in vitality and rumination with only 1-month of mindful walking in people with depressive disorder (*El-Sayed et al., 2024*).

After analysing the baseline variables, we categorised the participants as metabolically unhealthy, with overweight/obesity, according to some inflammatory biochemical parameters, overweight values, and low CRF (Table 1). The related clinical guidelines on cardiovascular disease prevention strongly recommend intensified attention and support to improve adherence to lifestyle changes in patients with mental disorders, including a reduction in sedentary time and a recommendation to follow general PA and exercise recommendations (*Visseren et al., 2021*). Based on our hypothesis, we expected that the exercise-mediated mitochondrial functions (*i.e.,* mitophagy) would increase CRF and improve physical and biochemical inflammation-related parameters and, consequently, the antidepressant effect (*Hird et al., 2024*; *Sun et al., 2022*). However, the lack of changes in the participants' metabolic profile and CRF (no improvement or worsening) requires further consideration of our results. First, the effect of low-to-moderate-intensity PA may have been amplified by reducing clinically related depression symptoms, although there were no changes in other parameters, consistent with previous studies (*Morga et al., 2021*; *Mota-Pereira et al., 2011*). Second, the intrinsic correlation between the pathophysiology of depression itself and inflammatory processes, with elevated CRP and IL-1β levels in the present study, mediated by the tryptophan-kynurenine pathway, makes the lack of favourable changes in the parameters related to body composition, CRF, and the biochemical profile very interesting. The maintenance of these parameters could be beneficial, considering the functional and clinical improvements (*Ren & Xiao, 2023*). Finally, high-intensity interval training seems to be more effective at inducing favourable changes to body composition, CRF, and inflammatory markers compared with low-to-moderate-intensity exercise, with exerkines, mitophagy, and lactate as the major brain-mediated effectors (*Heo, Noble & Call, 2023*; *Mishra & Thakur, 2023*).

Our study has several strengths, including the fact that there have been very few studies about the RMD population and none on an exercise intervention combining LIIT and resistance training. However, we must also note the limitations. First, we did not include a control group, which restricts our ability to control for confounding variables that might influence the observed outcomes. Moreover, we did not collect comprehensive data on potential covariates such as diet, sleep, or other lifestyle factors, which could impact physiological or psychological responses. These omissions were intentional, given the exploratory nature of this pilot study and the need to test feasibility with a limited sample size. Future studies should include randomised controlled designs and collect relevant covariate data to allow for adjusted analyses and stronger causal interpretations. Second, we did not have an equal number of men and women, which raises statistical problems. Future studies, particularly those using interventions and questionnaires, should seek to recruit an equal number of men and women. Third, four participants (22%) did not

complete the post-intervention assessments. While missing data can bias the results, we note that in this case, dropout was due to logistical factors, such as early hospital discharge and geographic distance, rather than lack of clinical benefit. Finally, the sample represented the RMD population from community mental health centres in a single city, where mental health and social care are highly regarded and may not be representative of other towns or communities worldwide.

## CONCLUSIONS

We demonstrated that a supervised, combined exercise programme (LIIT and resistance training) for people with RMD can induce positive and beneficial changes in functionality, clinical symptoms, and QoL, and maintain body composition, CRF, and biochemical variables. These results highlight the critical role of supervised exercise, regardless of intensity, in improving clinical symptoms in people with RMD. Thus, exercise might tentatively be considered in co-adjuvant programmes for the treatment of certain RMD populations.

## ACKNOWLEDGEMENTS

Thanks to Dr David V. Sheehan for the Sheehan Disability Scale license agreement.

### Funding

This study was funded by the "III Convocatoria Intramural de la Fundación Vital Fundazioa–IIS BIOARABA" and the Mental Health Network of Álava. The funders had no role in study design, data collection and analysis, decision to publish, or preparation of the manuscript.

### Grant Disclosures

The following grant information was disclosed by the authors:
'III Convocatoria Intramural de la Fundación Vital Fundazioa–IIS BIOARABA.
Mental Health Network of Álava.

### Competing Interests

The authors declare there are no competing interests.

### Author Contributions

- Mikel Tous-Espelosin conceived and designed the experiments, performed the experiments, analyzed the data, prepared figures and/or tables, authored or reviewed drafts of the article, and approved the final draft.
- Cristóbal Pavón-Navajas conceived and designed the experiments, performed the experiments, prepared figures and/or tables, and approved the final draft.
- José Etxaniz-Oses conceived and designed the experiments, performed the experiments, analyzed the data, prepared figures and/or tables, and approved the final draft.

- María Teresa Cañas-García performed the experiments, prepared figures and/or tables, and approved the final draft.
- Sara Maldonado-Martin conceived and designed the experiments, performed the experiments, analyzed the data, prepared figures and/or tables, authored or reviewed drafts of the article, and approved the final draft.
- Julene Orruño-Vivar performed the experiments, prepared figures and/or tables, and approved the final draft.
- Pablo Corres performed the experiments, analyzed the data, prepared figures and/or tables, and approved the final draft.
- Nagore Iriarte-Yoller conceived and designed the experiments, performed the experiments, prepared figures and/or tables, and approved the final draft.

## Human Ethics

The following information was supplied relating to ethical approvals (i.e., approving body and any reference numbers):

Ethics approval for this study was granted by the OSI Araba Research Ethics Committee on 7 May 2021, with Certificate No. 2021-045, following the Declaration of Helsinki.

## Clinical Trial Ethics

The following information was supplied relating to ethical approvals (i.e., approving body and any reference numbers):

This study was registered with the International Standard Randomized Controlled Trial Code NCT 05136027.

## Data Availability

The raw measurement are available in the Supplementary File 1. The raw data show the clinical trial participant consent form.

## Clinical Trial Registration

The following information was supplied regarding Clinical Trial registration:

NCT05136027.

## Supplemental Information

Supplemental information for this article can be found online at http://dx.doi.org/10.7717/peerj.20356#supplemental-information.

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
