# Peer review of "Effects of combined exercise training for adults with resistant major depression: a pilot study from the TRACE-RMD project"

_PeerJ, doi:10.7717/peerj.20356_

## Round 0.1 · original submission · Major Revisions

· Academic Editor

Major Revisions

Reviewer 1 has serious concerns about the study that must be addressed in detail.

**Language Note:** The review process has identified that the English language must be improved. PeerJ can provide language editing services - please contact us at [email protected] for pricing (be sure to provide your manuscript number and title). Alternatively, you should make your own arrangements to improve the language quality and provide details in your response letter. – PeerJ Staff

Reviewer 1 ·

Basic reporting

Throughout the manuscript, there were several areas where there should be prior literature referenced where it is not. For example, on line 66, the authors state that individuals generally practice an unhealthy lifestyle without citing appropriate literature. Another instance is within the method section where the authors state that Borg's scale was used, but then didn't reference that validation publication. These are just two of many examples throughout the manuscript.

In addition, there is a lack of sufficient introduction and background providing support as to why each of the dependent variables were chosen and why they would collectively advance this area of inquiry.

There is also a lack of citing the many meta-analyses and systematic reviews on the effects of exercise for depression, and the types of mediators/moderators that have been assessed.

Experimental design

As mentioned in box 3, some of the methods lack details to replicate, including the instructions given to participants before the assessments and the time of day some of these procedures were completed. Some of the screening procedures are also unclear as to who performed the screening procedures, and then how were the participants recruited to the study.

Validity of the findings

The authors concede that this study was not powered to detect significant effects, so then they state that it should be treated as a pilot study. This type of post-hoc explanation stands in opposition to the purpose of pre-registering a clinical trial. If this was indeed meant to be a pilot study, then the authors should focus on reporting effect sizes with uncertainty intervals, rather than still focus on changes in the dependent variables based on an alpha-threshold in the paired-samples t-tests.

It is unclear as to the validity of some of the findings, as the authors did not report psychometrics or validity coefficients of the instruments used to derive the dependent variables. In addition, some of the procedures have a lack of sufficient detail , which would impact some findings. For example, when was the bioelectrical impedance conducted? Were all these assessments at the same time of day at pre and post intervention? Were they at the same time of day for all participants?

Additional comments

The authors should be commended for pursuing this challenging yet important line of research. However, there is a lack of sufficient background and rationale for the selection of the dependent variables. This makes it unclear as to how the study contributes to the field beyond using a new exercise prescription but in an uncontrolled design.

Moreover, the statement, "Since the sample size was too small (n = 18) for adequate power for statistical significance, the current research was considered a pilot study," seems to suggest that the study was initially intended to be fully powered but later re-framed as a pilot due to a small sample size. This raises concerns about the interpretability of the findings and the meaningfulness of the statistical analyses carried out. Since the authors state this is a pilot study, it seems like only effect sizes should be reported; however, a conventional significance alpha threshold of .05 is still adopted. Therefore, the authors should consider whether an emphasis should be placed on significance or effect sizes. Regardless of this approach, it should be outlined and clear from the start of the manuscript.

·

Basic reporting

Overall, the manuscript used clear and professional language. The background and previous literature have been provided. The results are also presented based on the source tables/figures. Minor corrections needed for multiple sentences throughout the manuscript and some corrections for source figures.
1. Line 70-71: “Relatedly there is a bidirectional relationship…”
2. Line 84-108: During the literature review, the phrase “this population” has been used in multiple sentences to refer to the population of interest, but it is not clear which sentence defined “this population” and I am confused whether “this population” refers to patients with MDD, or RMD, or with disability, or elderly population. Please give more clarity on this phrase or specifically use “population with MDD/RMD/disability” instead of “this population”.
3. Line 98-100: It is hard to see how the referenced example is causally related to the previous text, which population does exercise seem to provide advantage? Please explain how the example justifies the statement.
4. Line 121: Since the number of women is 12 and the total number of participants is 18, the proportion of women participants should be 66.7% instead of 6.7%. Please correct this throughout the manuscript (including the abstract).
5. Line 136-138: The phrase “both in centimeters” could be moved to “Waist and hip circumferences were taken in centimeters”.
6. Line 160-162: Global comment on using numbered bullet points. It is not recommended to use “1)”, “2)”, etc. in a sentence. The sentence could be rephrased as “…with the presence of two or more of the following four criteria”.
7. Line 174 and line 195: Please clarify what “xxxx” means. I am not sure if this is a typo. If there is a specific Psychiatric Hospital that needs to be mentioned, please mention the name. Otherwise, if it is not important to mention the name, change the wording to be more general, such as “in a local psychiatric hospital”.
8. Line 206: Minor suggestion. The phrase “six intervals of 15 s at R2 interspersed with 1-min intervals at R1” could be improved. I understand this as repeating 15 s R2 followed by 1 min R1 for 6 times.
9. Line 228-237: This paragraph describes the results in Table 1. Please explain in the manuscript that Table 1 summaries for continuous measurements are in the format of mean+/-standard deviation, or median +/- IQR, or median +/- standard deviation, for scientific clarification. When describing the values, please mention they are the mean/median values.
10. Line 242: The phrase “for each group of participants” can be removed, the study only has one group.
11. Line 242-249: Same comment as No.9. For Table 2, please explain the format of the summaries.
12. Line 252-254: There is no source table/figure for the change of body composition variables and physiological variables and biochemical variables.
13. Line 254-261: Multiple issues with the reporting here. First, the Delta presented here is not absolute mean difference, but relative mean difference in percentage; however, the p-value for the paired sample t-test results from testing if the absolute mean difference is zero. Please explain why presenting the relative mean difference as Delta, and then clarify p-value is from paired sample t-test. Second, the Figure 3 has % as unit for the QoL scales, this is incorrect and please remove the unit for Figure 3. Also, please add legends to both Figure 2 and 3 to explain T0 and T1. Third, please double check the calculation of Delta after clarifying how it is calculated.

Experimental design

Overall, the experiment was well-designed, especially given that this is only a pilot study. However, I would like to give the suggestion below. As acknowledged by the authors in the discussion section, this study is a single arm study. Unlike a randomized study with control arm that can balance baseline characteristics to avoid potential confounders for the intervention, this study compared each participant at the times after intervention with himself/herself at baseline. Such self-controlled experiment typically requires a collection of all relevant covariates/independent variables that could affect the outcome (for example, information about participants nutrition plan, lifestyle, etc.) before and after the intervention, and adjust for any change in the covariates/independent variables in the analysis. However, given that the study is only a pilot study with a limited number of participants, it may not be practically useful to collect more covariates/independent variables. In this case, based on the current study design and data collection, the authors should be very careful in interpreting the analysis results.

Validity of the findings

Overall, the analysis and the results are clear. However, I have three main comments about the validity of the findings and recommend the authors to address them in the manuscript.
1. Related to the comment on reporting results from line 254 to 261, there is an inconsistency between presenting the relative mean difference (Delta) and the p-value for paired sample t-test (or Student’s t-test for repeated measures). The underlying hypothesis for the paired sample t-test is whether the absolute change of mean (after the intervention) equals to zero, it is not the same as testing whether the relative change of mean (in percentage) equals to zero. It is recommended to explain the relative mean difference (Delta) is only for presentation purpose, and please clarify how it was calculated.
2. One major drawback in the analysis is ignoring the 4 participants who dropped out of the study without post-intervention measurements. The total number of participants is 18, and 4 dropouts consist of a high proportion of missingness for the analysis data. Especially if the 4 participants dropped out due to no clinical improvement from the intervention, including these participants could significantly change the analysis results. For the manuscript purpose, the authors should at least explain the reasons for the dropouts. It is recommended that the authors perform certain sensitivity analyses to address the high proportion of dropouts in the pilot study, by properly handling the missing post-intervention data for the 4 dropouts.
3. It is recommended that the authors properly interpret the findings by explaining the assumption imposed for the analysis. The paired sample t-test was used for comparing the body composition, CRF, biochemical variables, clinical symptoms, and quality of life before and after the intervention, assuming no other potential confounders that could affect each of the outcomes during the time of intervention, so that the authors may claim the changes in the outcomes are due to the intervention.

---

## Round 0.2 · Minor Revisions

· Academic Editor

Minor Revisions

**Language Note:** When preparing your next revision, please ensure that your manuscript is reviewed either by a colleague who is proficient in English and familiar with the subject matter, or by a professional editing service. PeerJ offers language editing services; if you are interested, you may contact us at [email protected] for pricing details. Kindly include your manuscript number and title in your inquiry. – PeerJ Staff

Reviewer 1 ·

Basic reporting

-

Experimental design

-

Validity of the findings

-

Additional comments

I have no further comments at this time.

·

Basic reporting

The authors have revised most of the mistakes pointed out in the previous review. There are still some minor corrections needed on the revision for scientific clarity.

1. Regarding the presentation of confidence intervals in the Abstract and the Results sections. It is standard to present the confidence interval in the format of “lower limit - upper limit”, or “(lower limit, upper limit)”. The format seems incorrect in the current version, especially since the lower and upper limits are flipped and the signs are not clear. Please correct the format of confidence intervals throughout the manuscript.

2. Typos on revisions of certain words: it seems that the authors have changed words like “characterized”, “randomized”, “standardized”, etc., into “characterised”, “randomised”, “standardised”, and so on. This is fine, both versions are correct. However, during such revision, there are some typos in the revision. For example, Line 69 “scharacterised”, Line 97 “sprioritised”, etc. All have typos. The authors should double-check when revising. Additionally, I also spotted that Line 222 “twicea” should be corrected as “twice a”.

3. Regarding the Results section, Line 282-294 on presenting the pre-post changes. If the results present (absolute) mean difference, I suggest not using “MD” as an acronym because this may be confused with median difference, especially since the authors brought up the option of testing median difference using the Wilcoxon signed rank test in the statistical analysis sub-section. More comments on the statistical analysis sub-section can be found below. Additionally, if the authors decide to present units for certain variables in the Results section, please ensure that the point estimate of the mean difference and the interval estimate both have the same unit.

Experimental design

The authors have added additional clarifications on the study design and data collection in the discussion section. The authors have also updated the interpretation of the findings. These updates are considered to be sufficient.

Validity of the findings

The authors have addressed the previous comments in the manuscript. However, I saw some additional ambiguities in the revision of the statistical analysis sub-section that need to be clarified; the authors need to revise the language or maybe certain results if they decide to follow the revised analysis workflow.

1. Line 245-251: The authors need to explicitly clarify that these analyses (test normality and summarize descriptively) are only about baseline values for each of the variables. Per my understanding, the results in Table 1 are only about baseline values of the variables, so I assume the authors tested the normality for baseline values of the variables and descriptively summarized the baseline values.

2. Line 250-251: just a minor comment, for non-normally distributed variables, the format to present their baseline descriptive summaries should be median +/- IQR as shown in Table 1.

3. Line 251-253: This part should be separated from the above analyses for baseline values, as it is about the pre-post changes for each of the variables. Please note that the authors should then test the normality for the pre-post difference to decide whether to use a paired t-test or a Wilcoxon signed rank test. Testing the normality of baseline values (to decide whether to summarize in mean +/ SD or median +/- IQR) and testing the normality of the pre-post difference (to decide whether to use a paired t-test or a Wilcoxon signed rank test for the significance of the pre-post difference) should be separated and clarified explicitly.

4. Line 253-254: It is good that the authors considered presenting the confidence intervals on the mean difference. However, now that the authors brought up the option of the Wilcoxon signed rank test for non-normal pre-post differences (which was not introduced in the previous manuscript), the authors should clarify that this method only gives the confidence interval on the median difference if assuming the distribution of the difference is symmetric. It is not common to present confidence intervals on median difference if one decides to use the Wilcoxon signed rank test, and I am not sure if the Wilcoxon signed rank test was used, since the results seemed to be the same as the previous ones. But if the authors did use the Wilcoxon signed rank test and obtained a confidence interval on median difference in the software, they should further clarify that the confidence interval is on median difference, assuming the distribution of the difference is symmetric.

Additional comments

In summary, there are still some minor corrections needed for the language and results in the manuscript after the revision. Also, with the update in the statistical analysis, the authors need to provide additional clarification on the analysis workflow. This may be only a technical issue to clarify and does not necessarily change the conclusion overall.

---

## Round 0.3 · accepted · Accept

· Academic Editor

Accept

Thank you for your efforts in addressing the reviewer comments. Your manuscript is now accepted for publication.

·

Basic reporting

The authors have addressed the comments regarding the format for confidence intervals, typos due to track changes, and the use of the acronym in the Results section.

Experimental design

No additional comment on the experimental design since the previous revision has been sufficient.

Validity of the findings

The authors have made substantial revisions to the text about the statistical analysis workflow and have provided a detailed description of the analyses planned and carried out in the statistical analysis section and the Results section, respectively.

I am satisfied with the language in interpreting the analysis results, just to provide advice to the authors regarding the text in Lines 287-295 that can improve the scientific clarity. The statements in Lines 287-295 are directional (e.g., significant reduction/increase), unlike the statements in Lines 282-287 (significant changes). Please note that if the authors want to make a directional conclusion, the corresponding statistical tests need to be one-sided tests instead of two-sided tests. If the authors use two-sided tests and want to make a conclusion, one can rephrase the wording to be, as an example, "there was a significant change in certain characteristics with a decreasing/increasing trend", instead of using phrases like "significant reduction/increase". This point is very minor, and readers should be able to comprehend.